# Multi-scale modelling of location- and frequency-dependent synaptic plasticity induced by repetitive magnetic stimulation in the dendrites of pyramidal neurons

Nicholas Hananeia[1,2]*, Christian Ebner[1,2,3], Christos Galanis[4,5,6,7], Hermann Cuntz[1,2,8,9], Alexander Opitz[10], Andreas Vlachos[4,5,6,7], Peter Jedlicka[1,2]

**1** Computer-Based Modelling in the Field of R Animal Protection, Faculty of Medicine, Justus Liebig University Giessen, Giessen, Germany, **2** Translational Neuroscience Network Giessen, Giessen, Germany, **3** Charité · NeuroCure (NCRC), Charité Universitätsmedizin Berlin, **4** Department of Neuroanatomy, Institute of Anatomy and Cell Biology, Faculty of Medicine, University of Freiburg, Freiburg, Germany, **5** BrainLinks-BrainTools Center, University of Freiburg, Freiburg, Germany, **6** Bernstein Center Freiburg, University of Freiburg, Freiburg, Germany, **7** Center for Basics in Neuromodulation (NeuroModulBasics), Faculty of Medicine, University of Freiburg, Freiburg, Germany, **8** Ernst Strüngmann Institute (ESI) for Neuroscience in Cooperation with the Max Planck Society, Frankfurt am Main, Germany, **9** Frankfurt Institute for Advanced Studies, Frankfurt am Main, Germany, **10** Department of Biomedical Engineering, University of Minnesota, Minneapolis, Minnesota, United States of America

\* nickhananeia@gmail.com

## Abstract

**Background:** Repetitive transcranial magnetic stimulation (rTMS) induces long-term changes in synapses, but the mechanisms behind these modifications are not fully understood. Although there has been progress in the development of multi-scale modeling tools, no comprehensive module for simulating rTMS-induced synaptic plasticity in biophysically realistic neurons exists.

**Objective:** We developed a modelling framework that allows the replication and detailed prediction of long-term changes of excitatory synapses in neurons stimulated by rTMS.

**Methods:** We implemented a voltage-dependent plasticity model that has been previously established for simulating frequency-, time-, and compartment-dependent spatio-temporal changes of excitatory synapses in neuronal dendrites. The plasticity model can be incorporated into biophysical neuronal models and coupled to electrical field simulations.

**Results:** We show that the plasticity modelling framework replicates long-term potentiation (LTP)-like plasticity in hippocampal CA1 pyramidal cells evoked by 10-Hz repetitive magnetic stimulation (rMS). In line with previous experimental studies, this plasticity was strongly distance dependent and localised to the proximal synapses of the neuron. We predicted a decrease in the plasticity amplitude for 5 Hz and 1 Hz protocols with decreasing frequency. Finally, we successfully modelled plasticity in distal synapses upon local electrical theta-burst stimulation (TBS) and predicted

**Data availability statement:** Code/data available on Zenodo: https://zenodo.org/records/17434411.

**Funding:** This work was supported by the Federal Ministry of Education and Research, Germany (BMBF, 01GQ2205B to PJ, 01GQ2205A to AV) and by NIH (R01EB034143 and R01NS109498 to AO). The funders had no role in study design, data collection and analysis, decision to publish, or preparation of the manuscript.

**Competing interests:** The authors have declared that no competing interests exist.

proximal and distal plasticity for rMS TBS. Notably, the rMS TBS-evoked synaptic plasticity exhibited robust facilitation by dendritic spikes and low sensitivity to inhibitory suppression.

**Conclusion:** The plasticity modelling framework enables precise simulations of LTP-like cellular effects with high spatio-temporal resolution, enhancing the efficiency of parameter screening and the development of plasticity-inducing rTMS protocols.

## Author summary

It is crucial to better understand how transcranial magnetic stimulation (TMS) leads to lasting changes in the strength of synaptic connections. While previous research has shown that TMS can induce long-term synaptic plasticity, the underlying mechanisms are still unclear, and current computer models don't fully capture how these changes happen at the level of individual cells. To address this, we built a modeling tool that simulates how different patterns of stimulation affect synaptic strength. We tested our model on plasticity data from hippocampal CA1 pyramidal cells. We found that it successfully reproduced results from experiments in organotypic slice cultures, particularly the increase in the strength of excitatory synapses close to the cell body induced by 900 pulses of 10 Hz magnetic stimulation. We also investigated the effects of different frequencies of stimulation and bursting stimulation, and found that higher frequencies cause greater increase in synaptic strength, and bursting stimuli can also increase synaptic strength at distal dendritic synapses.

## In brief

Here we develop a validated framework designed for simulating long-term synaptic plasticity induced by TMS. This framework offers accurate predictions of layer-specific effects of rTMS-induced synaptic changes and streamlines the screening of rTMS parameters for inducing synaptic plasticity.

## Highlights

- First rigorously validated model of TMS-induced long-term synaptic plasticity in extended neuronal dendrites that goes beyond point-neuron and mean-field modelling
- Robust simulations of experimental data on LTP-like plasticity in the proximal dendrites of CA1 hippocampal pyramidal cells evoked by 10 Hz repetitive magnetic stimulation (rMS)
- Replication of distal synaptic plasticity for a local electrical theta burst stimulation (TBS) protocol
- Prediction of distal and proximal LTP-like plasticity for rMS TBS
- 1 Hz rMS does not induce long-term depression

# Introduction

Transcranial magnetic stimulation (TMS, [1]) is a widely used non-invasive method of brain stimulation that has been shown to be effective in treating a variety of neurological conditions [2], including stroke [3], pain and drug-resistant depression [4–6]. Additionally, TMS has been investigated as a therapeutic intervention for tinnitus, epilepsy, Alzheimer's disease [7], Parkinson's disease [8], schizophrenia [9] and other conditions. Despite its widespread clinical and research applications, there are still significant gaps in our understanding of the mechanisms by which TMS exerts its effects [10, 11]. As its use continues to expand, it is increasingly important to better understand the effects of TMS at the whole-brain, network, but also at the cellular level [12,13].

TMS induces strong electric fields in the brain tissue beneath the stimulation coil, leading to neuronal activation. In particular, the electric field can elicit action potentials that originate predominantly in the axonal arbors of stimulated neurons [13,14]. The clinical efficacy of TMS is thought to arise primarily from its modulation of neuroplasticity [15]. However, the links between the TMS-induced electric fields, neuronal spiking, and long-term synaptic plasticity remain incompletely understood.

Experimental studies have demonstrated a range of TMS effects on the target area. In the hippocampal CA1 region, for example, TMS has been shown to increase intrinsic excitability [16], and to induce or modulate LTP [17–26] (see also [27–29]). Importantly, TMS induces electric fields affecting the entire neuronal morphology, which differs from classical LTP induction protocols that rely on focal direct electric stimulation of axons[28]. To model LTP induced by TMS in a bio-physically realistic manner, we employed a previously established computational model of synaptic plasticity [30]. This model is particularly suited for simulating changes in synaptic strength following repetitive magnetic stimulation (rMS) as it generates detailed spatiotemporal profiles of plasticity resulting from local compartment-specific voltage events such as dendritic spikes, NMDA spikes, large EPSPs, back-propagating somatic spikes, dendritic plateau potentials or combinations thereof [30].

Experimental studies using rMS have reported that LTP is preferentially induced in synapses located in the proximal dendrites [31] with LTP amplitude decreasing as a function of distance from the soma [31,32]. In this study, we were able to reproduce this distance-dependent pattern of rMS-induced LTP in our computational model. Moreover, our results predict that LTP amplitude depends on the rMS frequency and that rMS can also induce LTP in distal tuft synapses when a theta-burst stimulation protocol (rMS TBS) is applied.

# Materials and methods

## Simulation software and code availability

The code for the model generator as well as example simulations for all of the results in the results section is available here:

https://zenodo.org/records/17434411

All *NEURON* simulations were performed using the 7.8.2 release build of *NEURON* [33]. *NeMo-TMS* [34] was run on a modified version of the latest release build (2.0) on *MATLAB R2019b*. *HippoUnit* was run using *Python 3.9*. All simulations and validations were performed on a Windows 10 PC with an Intel Core i9-9900k processor running at 4.7 GHz and 48 GB of RAM.

For each simulation set, we generated a set of ten models to show the spread of simulation data. To do this, we randomised the synapse locations while keeping the total synaptic input current the same. The same set of ten models was used for each simulation set to enable direct comparison.

## Computational model of CA1 pyramidal cell

For all simulations here, we implemented a modified form of a biophysically complex but morphologically reduced model [35] of a CA1 pyramidal cell in the NEURON simulation environment. The reduced model has the branching structure

typical of a pyramidal cell, illustrated in Fig 1. It specifically has two basal dendrites at distances corresponding to the stratum oriens, each with two branches. In addition, it has an apical dendrite with a thickened trunk that branches into two distal tuft branches at a distance that corresponds to the stratum lacunosum-moleculare. At distances which would be associated with the stratum radiatum, there are three oblique branches which connect directly to the apical trunk.

The biophysical model has the following ion channels, which vary based on region, in addition to the default passive properties. In the axon, there are Na$^+$ channels and A-type, M-type, and delayed rectifier K$^+$ channels. In the dendrites, there are Na$^+$ channels, A-type, M-type, and delayed rectifier K$^+$ channels, and HCN (Hyperpolarization-activated cyclic nucleotide–gated) or I$_h$ channels, L-type, T-type, and N-type Ca$^{2+}$ channels, and calcium-gated K$^+$ channels. The soma contains all of the channels present in the dendrites, with the addition of a M-type K$^+$ channel. The conductances for all of these channels vary based on location; these are specified in Table 1.

The axon consisted of an axon initial segment and an unmyelinated terminal segment as well as 5 nodes of Ranvier and 6 myelinated inter-nodal segments, as myelin is necessary to facilitate the firing of the cell in response to TMS [14,34]. The axonal initiation of the magnetic stimulation-initiated action potential is shown in S6 Fig. A schematic of the axon morphology is shown in S7 Fig.

To ensure that the spiking characteristics were in accordance with realistic behaviour for a CA1 pyramidal cell, the axon initial segment had an elevated sodium conductance as well as elevated K$_A$ and K$_M$ conductances [36]. The values of these conductances were derived by increasing them to a sufficient level where the smaller axon initial segment generated similar electrophysiological properties to the original model with the *HippoUnit* validation suite (see below, [35]).

A list of the specific channels, their conductances, and the passive properties of the different cell compartments is given in Table 1, with properties that have been modified from the original CA1 pyramidal cell model in bold italics. Three

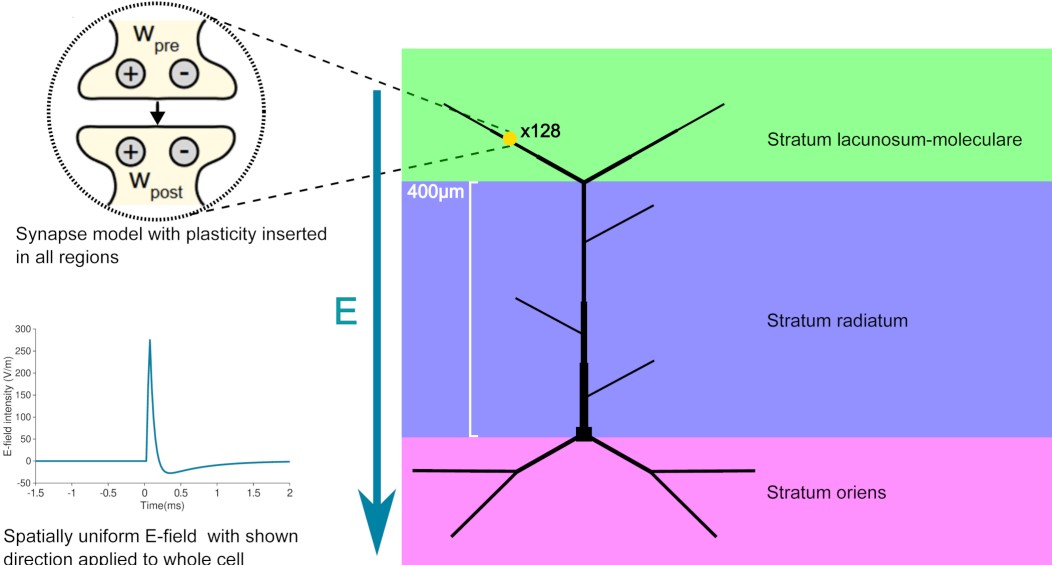

**Fig 1. Modelling framework for TMS-induced long-term synaptic plasticity.** The Neuron Modeling for TMS (*NeMo-TMS*) toolbox integrates detailed neuronal models with TMS-induced electric fields, allowing the simulation of cellular and subcellular voltage and calcium responses during single and repetitive TMS pulses [34]. The direction of the electric field is represented by the vector E. We implement in *NeMo-TMS* a validated model of a CA1 pyramidal cell with detailed biophysics and reduced morphology, capable of generating realistic dendritic and somatic spikes [35,43]. Relative dendritic diameters are depicted. We introduced a unified voltage-dependent 4-pathway (pre- and postsynaptic; LTP and LTD) model of long-term synaptic plasticity (yellow circle with pre- and postsynaptic LTP (+) and LTD(-)), capable of reproducing the frequency-, timing- and location-dependence of synaptic changes [30], into the existing *NeMo-TMS* framework. 128 of plastic excitatory synapses were placed in the morphology.

**Table 1. Ion channels and membrane properties.** Changes from the original [35] are shown in bold italics.

| Property | Soma | Apical trunk | Dendrites | AIS and nodes | Myelin | Axon terminal |
|---|---|---|---|---|---|---|
| $Na$ | 0.035 | ***0.015-0.02*** | 0.03828 | ***0.5*** | | 0.035 |
| $K_A$ | 0.0075 | 0.1-0.2 | Variable | ***0.44*** | | 0.0163694 |
| $K_{DR}$ | 0.0015 | 0.02 | 0.004304 | ***0.04*** | | 0.011664 |
| $K_M$ | 0.001 | | | ***0.02*** | | 0.0264739 |
| $K_{Ca}$ | 0.0015 | 9.031e-5 | 9.031e-5 | | | |
| $Cag-K$ | 4.48e-5 | 4.48e-5 | 4.48e-5 | | | |
| $Ca_T$ | 0.00005 | 1.185e-6 | 1.185e-6 | | | |
| $Ca_L$ | 0.0005 | 8.033e-6 | 8.033e-6 | | | |
| $Ca_N$ | 2.26e-6 | 2.26e-6 | 2.26e-6 | | | |
| $I_h$ | Variable | Variable | Variable | | | |
| $C_M$ | 1 | 1 | 1 | 1 | ***0.01*** | 1 |
| $R_A$ | 115 | 115 | 115 | 85.2 | ***60*** | 85.2 |
| $g_{pas}$ | 9.031e-5 | 9.031e-5 | 9.031e-5 | 1.29e-4 | ***8.89e-7*** | 1.29e-4 |
| $e_{pas}$ | Variable | Variable | Variable | -79.92 | -79.92 | -79.92 |
| **Spatially variable property** | **Equation; $d$ is distance from soma in $\mu m$** | | | | | |
| $I_h$ | $1.904e-5\frac{1+3d}{100}$ | | | | | |
| $K_A$ | $0.0129\frac{15}{1+exp(\frac{300-d}{50})}$ | | | | | |
| $e_{pas}$ | $-65.73-\frac{5d}{150}$ | | | | | |

properties were ranged within their given regions and the equations governing their values are given in Table 1. None of these ranged variables have been changed from the original model.

## Validation of the CA1 pyramidal cell model using *HippoUnit* tests

Because the reduced model of the CA1 cell was augmented with an axon and a realistic axon initial segment to enable it to fire in response to TMS pulses, in addition to importing the model into the *NeMo-TMS* framework, re-validation using the *HippoUnit* test suite was required to ensure that our modified model still passed the HippoUnit tests with an acceptable error margin.

All five *HippoUnit* tests [35,37] were performed - the somatic features test, the back-propagating action potential test, the depolarisation block test, the PSP attenuation test and the oblique integration test. These tests are described in detail in Tomko et al. [35] and no changes were made to the test settings used for validation in that paper. The resulting final error scores are shown in S8 Fig.

## Synaptic models

The excitatory synaptic mechanism used in the plasticity simulations for AMPA/NMDA synapses was the voltage-dependent 4-pathway model of Ebner et al. [30], while for inhibitory synapses the non-plastic double exponential synapse Exp2Syn mechanism, with a negative reversal potential of -75 mV was used.

The Ebner plasticity model uses a group of internal variables to model the state of the synapse. The most important ones are those that represent critical points in the four LTP/LTD pathways that it models. Notably, postsynaptic LTP and LTD are governed by $C$ which is loosely related to calcium entering NMDARs. Presynaptic LTD is gated by $T$, loosely representing voltage-gated calcium channels, while presynaptic LTP is determined by $N_\alpha$ and $N_\beta$, which could be related to L-type calcium channels and the nitric oxide processes respectively.

The full extent of the pathway structure and their dynamics is illustrated in Fig 1 of Ebner et al (2019) [30].

These four pathways contribute to the final weight change at each time step, with each internal state variable having different rise/decay courses. At every time step, the contribution for each of the four pathways (presynaptic LTP, presynaptic LTD, postsynaptic LTP, postsynaptic LTD) is additively combined into two (presynaptic and postsynaptic) contributions. The current total weight of the synapse is then calculated as the product of presynaptic and postsynaptic weights.

The default parameters for this model were tuned to neocortical plasticity data and were not appropriate for a simulation with a larger number of hippocampal synapses. Apart from the parameters in the Ebner model which were derived from direct biophysical measurements, free parameters such as LTP/LTD amplitudes and thresholds were tuned by hand to match the experimental results from Ikegaya et al. [38]. Initialised synaptic weights for the excitatory synapses were uniformly random within a given interval; this interval was manually tuned as for the other parameters. Synaptic weights were chosen with a multiplier between 0.22 and 0.44 (the tuned interval). This was multiplied by the default initial weight factors of 0.5 for presynaptic weights and 2 for postsynaptic weights. For all simulations except where otherwise noted, 128 excitatory synapses and 18 inhibitory synapses were placed in relative numbers according to Megias (2001) [39]. A table of parameters for the synaptic plasticity model is shown in Table 2.

To examine the effect of different synapse locations on the plasticity, we generated a population of different sets of synapse locations for each simulation. This was done by adjusting the seed of the random number generator. The total input synaptic current for each of these models was kept constant; the choice of weights was the same for each model. The same sets of 10 synapse locations were used for all group results shown, to enable direct comparison between the effects of different stimulation protocols.

## Pharmacology *in silico*

To simulate pharmacological perturbations of the model, the following parameter modifications were made to the model. To simulate bicuculline, the weight of all inhibitory synapses was set to zero. To simulate dendritic TTX application, the conductances of all sodium channels in the dendrites of the cell were set to zero. Similarly, to simulate L-type calcium channel block and total calcium channel block, the conductance of these channels in the cell was set to zero.

As the plasticity model is based on NMDA receptor and calcium dependent mechanisms, the NMDA receptor blockade and calcium channel blockade required additional modifications to the synaptic plasticity model itself. For L-type calcium

**Table 2**. Parameter values for four-pathway synaptic plasticity model.

| Parameter | Definition | Value | Parameter | Definition | Value |
|---|---|---|---|---|---|
| $W_{MIN}$ | Min. weight scale factor | 0.22 | $M_T$ | $T$ saturation function slope | 1.7 |
| $W_{MAX}$ | Max. weight scale factor | 0.44 | $\theta_U^N$ (mV) | $u$ to $N$ voltage threshold | -55 |
| $\tau^A$ (ms) | AMPA current rise time | 0.2 | $\tau_Z^A$ (ms) | $Z$ function rise time | 1 |
| $\tau^B$ (ms) | AMPA current decay time | 2 | $\tau_Z^B$ (ms) | $Z$ function decay time | 15 |
| $e_{Na}$ (mV) | Reversal potential | 0 | $M_Z$ | $Z$ saturation function slope | 6 |
| $S_{AMPA}$ | AMPA current contribution | 0.5 | $\tau_{N\alpha}$ (ms) | $N_\alpha$ time constant | 7.5 |
| $S_{NMDA}$ | NMDA current contribution | 0.5 | $\tau_{N\beta}$ (ms) | $N_\beta$ time constant | 20 |
| $\tau_G^A$ (ms) | NMDA current rise time and $G$ rise time constant | 2 | $M_{N\alpha}$ | $N_\alpha$ saturation function slope | 2 |
| $\tau_G^B$ (ms) | NMDA current decay time and $G$ decay time constant | 50 | $M_{N\beta}$ | $N_\beta$ saturation function slope | 10 |
| $W_{pre}^{init}$ | Initial presynaptic weight | 0.5 | $\theta_N^X$ | $N$ to $X$ activation threshold | 0.3 |
| $W_{post}^{init}$ | Initial postsynaptic weight | 2.0 | $\theta_u^C$ (mV) | $u$ to $C$ voltage threshold | -63.5 |
| $M_G$ | $G$ saturation function slope | 10 | $\theta_C^-$ | Postsynaptic LTD threshold | 8.0 |
| $A_{pre}^+$ | Scaling factor on presynaptic LTP amplitude | 1.4e-5 | $\theta_C^+$ | Postsynaptic LTP threshold | 8.2 |
| $A_{pre}^-$ | Scaling factor on presynaptic LTD amplitude | 2.25e-4 | $\tau_{K\alpha}$ (ms) | $K_\alpha$ time constant | 10 |
| $A_{post}^+$ | Scaling factor on postsynaptic LTP amplitude | 1.6e-2 | $\tau_{K\gamma}$ (ms) | $K_\gamma$ time constant | 25 |
| $A_{post}^-$ | Scaling factor on postsynaptic LTD amplitude | 45e-5 | $M_{K\alpha}$ | $K_\alpha$ saturation function slope | 1.5 |
| $\tau_u^T$ (ms) | Filtering constant from $u$ to $T$ | 10 | $M_{K\beta}$ | $K_\beta$ saturation function slope | 1.7 |
| $\theta_u^T$ (mV) | $u$ to $T$ voltage threshold | -60 | $S_{K\beta}$ | $K_\beta$ scale factor | 100 |

channel blockade, we set the internal variable $N_\alpha$ to zero at each time step, corresponding to the L-type calcium component of presynaptic LTP. NMDA receptor blockade was simulated by setting the parameter $S\_NMDA$, representing the NMDA fraction of the combined AMPA-NMDA synapse model to zero (thereby removing the synaptic NMDA component, and correspondingly reducing maximum synaptic conductance), as well as forcing the internal variable $C$, representing NMDA receptor activation, to zero at each time step (thereby disabling the internal synaptic plasticity component associated with NMDA activation). As NMDA blockade would also eliminate the postsynaptic calcium influx, we also set the variable $N_\alpha$ to zero at each time step. To simulate a calcium-free solution, all calcium influx mechanisms were disabled, as well as calcium-dependent variables within the synaptic model. With the exception of the NMDA blockade removing the NMDA component of the postsynaptic potential, none of these perturbations modified the synaptic conductance.

## Stimulation protocols

To simulate rMS, we use the same implementation as present in the NeMo-TMS [34] framework, where the magnetic stimulation is implemented as an extracellular voltage source coupled to the compartmental neuronal model via the extracellular voltage variable in NEURON. In this case, we use the uniform electric field approximation included with NeMo-TMS, which is coupled to every compartment in the morphology at runtime.

This is done by the following approximation, where $V_e$ is the extracellular voltage, $E$ is the electric field vector, and $ds$ is the differential of a given segment of membrane [14], and $E_{(x,y,z)}$ are the x, y, and z components of the electric field at location $(x,y,z)$:

$$V_e(x, y, z) = - \int \vec{E} \cdot d\vec{s} = -\vec{E} \cdot \vec{s} = -(E_x x + E_y y + E_z z) \tag{1}$$

This accounts for the spatial component; the temporal component is encoded in a separate timecourse which is multiplied by this uniform spatial component; when there is no active extracellular voltage applied when the time course is zero; the time course also contains the temporal shape (monophasic or biphasic) of the stimulus. Thus, when the stimulus is applied, every location (every postsynaptic compartment) in the cell receives extracellular stimulus. Importantly, although we used a uniform electric field, NeMo-TMS allows the user to apply any (spatially and temporally complex) extracellular field simulated by SimNIBS (https://simnibs.github.io/simnibs/build/html/index.html).

When simulating rMS pulses, we also stimulated all (both excitatory and inhibitory) synapses simultaneously with the application of the rMS pulse to the entire cell. This decision was made based on the assumption that an rMS pulse will cause all (or a large sub-population of) the cells in the target area to fire simultaneously. Given the close proximity of CA1 and CA3, as well as the likelihood that action potentials will be induced in the terminals of axons in the target area without activation of their cell bodies [14], the assumption of near-simultaneous activation of all synapses was implemented in our simulations. To account for the delay in synapse activation, we apply a 1ms delay from the onset of the magnetic stimulus to postsynaptic activation.

In addition to the rMS pulses, we applied a Poisson input at 3 Hz to all synapses mimicking spontaneous neuronal activity in organotypic slice cultures.

For the temporal component of the stimulation, the TMS waveform generator supplied with *NeMo-TMS* was used to generate a monophasic waveform with an amplitude 275 V/m, which was 60 V/m above the minimum required to induce an action potential, which was synchronised with the presynaptic activation of all synapses, with this synaptic activation delayed 1 ms after the magnetic stimulus to simulate synaptic activation delay.

Likewise, theta burst stimulation simulations utilised a 250 V/m biphasic waveform with synchronous activation of all synapses. The theta burst stimulation protocol consisted of 15 bursts of 5 stimuli at 100 Hz, with each burst separated by 100 ms.

Local electrical stimulation represented the classical stimulation of axons by electrodes and was simulated by synchronous activation of all synapses in the target region at the given frequency.

## Results

### A module for simulating TMS-induced long-term synaptic plasticity

Existing modelling tools for TMS integrate anatomically and biophysically detailed neuronal models with simulations of the electric fields induced by TMS. This allows for the simulation of cellular and subcellular voltage and calcium dynamics in response to single and repetitive TMS pulses [14,34,40–42]. However, these tools lack experimentally validated plasticity rules necessary for simulating standard LTP induction protocols. To bridge this gap, we have extended the Neuron Modeling for TMS (*NeMo-TMS*) toolbox (Fig 1) with a comprehensive synaptic plasticity model that incorporates a robust four-pathway mechanism of synaptic modifications [30]. The synaptic plasticity model we use here has been extensively validated in our previous work [30], accurately capturing a range of observed plasticity phenomena, including spike-timing-dependent plasticity (STDP) induced by various pairing protocols, dendritic spike-induced plasticity, and subthreshold event-induced plasticity [30]. Our goal was to enable the simulation of long-term synaptic changes triggered not only by classical electrical synaptic stimulation but also magnetic stimulation (see Methods). We integrated the plasticity model with a reduced model of CA1 pyramidal neurons (Fig 1) that has been validated [35] in a comprehensive testing environment called *HippoUnit* [35,37,43]. This CA1 pyramidal cell model effectively captures five key electrophysiological properties including somatic spiking, back-propagating action potentials, and dendritic integration of synaptic inputs [35], maintaining a balance between detailed biophysical mechanisms and computational tractability. This model facilitates the exploration of various LTP/LTD scenarios under rTMS, providing insights into the specific changes across different anatomical layers and synaptic inputs.

### Validation of the synaptic plasticity model for LTP/LTD induced by local electrical stimulation

Our main aim was to develop a plasticity modelling framework that would reproduce and explain our previously acquired data on rMS-induced changes of synapses in CA1 pyramidal cells from organotypic slice cultures [31,44]. However, the unified voltage-dependent model of synaptic plasticity we chose to use was previously tuned to several predominantly neocortical but not hippocampal plasticity datasets [30]. Therefore, we first tested whether the plasticity model could reproduce synaptic plasticity data from classical hippocampal CA3-CA1 synapses.

As a reference hippocampal dataset, we used the frequency-dependent synaptic weight changes from Ikegaya et al. [38], resulting from the induction of LTP or LTD by local extracellular electrical stimulation. In these experiments, the induced LTP/LTD was measured both in the absence of pharmacological perturbation of the culture, and with the application of bicuculline, a $GABA_A$ antagonist that suppresses GABAergic inhibition. To simulate the split between control and bicuculline results, we either enabled or disabled the set of inhibitory synapses (see Methods), with the bicuculline case simulated with zero inhibitory synapses enabled. As in Ikegaya et al. [38], the stimulation protocol was a series of 900 pulses delivered to the Schaffer collaterals at a constant frequency of either 1, 10, 30, or 100 Hz. Inhibitory synapses also received an identical stimulus for the simulations where inhibition was active. As in our other simulations, there was an additional 3 Hz Poisson random background activity in all synapses to represent the non-silent nature of slice cultures.

After adjusting the parameters of the plasticity model (shown in Table 2), our model produced results within standard deviation for all conditions except for the 1 Hz control case and the 10 Hz inhibition case (Fig 2). In agreement with the data, significant LTP was only induced above 30 Hz. The change in synaptic weights was negligible at frequencies below 10 Hz, a condition that will be compared later with the rMS results.

To measure goodness of fit, we calculated a root mean squared error, normalised by the SEM of the data. For the inhibition enabled case, these normalised errors were, for 1, 10, 30, and 100 Hz respectively, 4.45, 3.68, 1.88, and 0.9. Similarly, for the inhibition blockade case, these errors were 0.36, 0.75, 0.1, and 0.6. Thus, our model shows the best fit for high frequency (≥30) Hz cases and the worst fit for the low frequency cases with inhibition blocked.

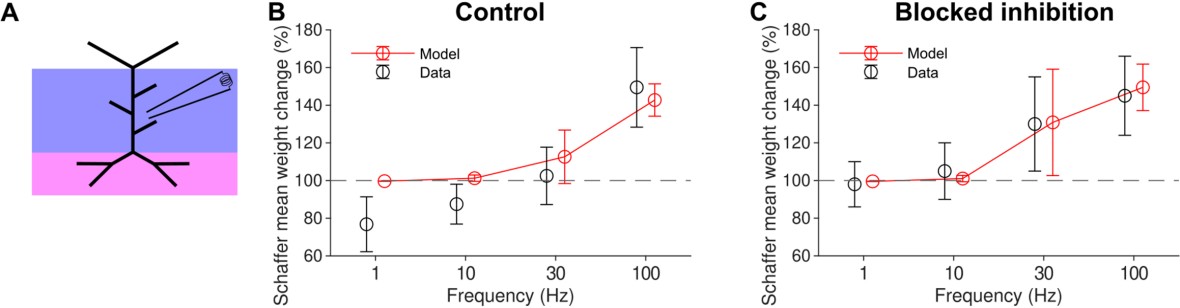

**Fig 2**. **Unifying four-pathway model of long-term synaptic plasticity reproduces frequency dependence of synaptic changes in hippocampal CA1 pyramidal cells evoked by local electrical Schaffer collateral stimulation** *(a)* Local electrical stimulation (i.e. Schaffer collateral synaptic stimulation) regions (blue purple and magenta) shown on schematic. There is no somatic stimulation. *(b)* Schaffer collateral weight change for 900 pulses delivered at 1, 10, 30, and 100 Hz with inhibition active. Model (red) and data (black). *(c)* Same as b but with inhibition blocked. Model (red) and bicuculline data (black).

These results indicate that our model could reproduce realistic LTP amplitudes for local electrical stimulation of hippocampal CA1 pyramidal cell synapses. Therefore, the parameters that generated these results (see Table 2) were used for all subsequent simulations.

To simulate an additional synaptic plasticity paradigm, we use data from acute hippocampal slice recordings in young adult rats [45]. Our model also successfully replicated a burst plasticity induction protocol, in which pairing presynaptic events with single postsynaptic events does not induce LTP, but pairing presynaptic events with postsynaptic bursts does induce significant LTP (S5 Fig). Furthermore, as mentioned above, our plasticity model [30] has previously been shown to reproduce additional cortical synaptic plasticity paradigms such as pairing protocols, dendritic spike-induced plasticity and subthreshold event-induced plasticity.

### Synaptic plasticity model reproduces and explains experimentally observed proximal LTP induction by rMS

To determine whether our plasticity model would be applicable to a realistic rMS protocol, we aimed to simulate the plasticity data of Lenz et al. [31]. In this study, an rMS of 900 pulses at 10 Hz was applied to hippocampal slice cultures, inducing strong LTP in the proximal stratum radiatum of CA1. We modelled this with a train of 900 of *NeMo-TMS*'s monophasic pulses with an amplitude set to 275 V/m, clearly above the minimum firing threshold of the CA1 pyramidal cell model. Each *NeMo-TMS* pulse was synchronised with the simultaneous activation of all synapses, because we assumed that rMS simultaneously activates not only the CA1 pyramidal cell axon but also the presynaptic axons.

As a result, the mean Schaffer collateral synaptic weight increased by 30%, with the LTP induction strongly localised in the synapses of the proximal dendrites. This was consistent with experimental observation of selective potentiation of proximal synapses in CA1 pyramidal cell dendrites following 10 Hz rMS [31]. Considering only the proximal synapses of the stratum radiatum (defined here as those in the first half of the apical trunk and associated obliques), an average LTP of 65% above baseline (direct weight value comparison from pre-stimulus) was achieved, whereas the distal synapses of the stratum radiatum showed no LTP (Fig 3a).

To predict the frequency dependence of synaptic plasticity induced by the rMS protocol, we delivered the stimuli at four additional frequencies of 1, 5, 6, 8, and 9 Hz. Proximal LTP was induced, (Fig 3b), with the amount of potentiation increasing with frequency.

Consistent with data, our simulations (Fig 3) showed that rMS was able to induce significant LTP at lower frequencies than the commonly used presynaptic plasticity-inducing stimulation protocols (simulated in Fig 2). Experiments show that

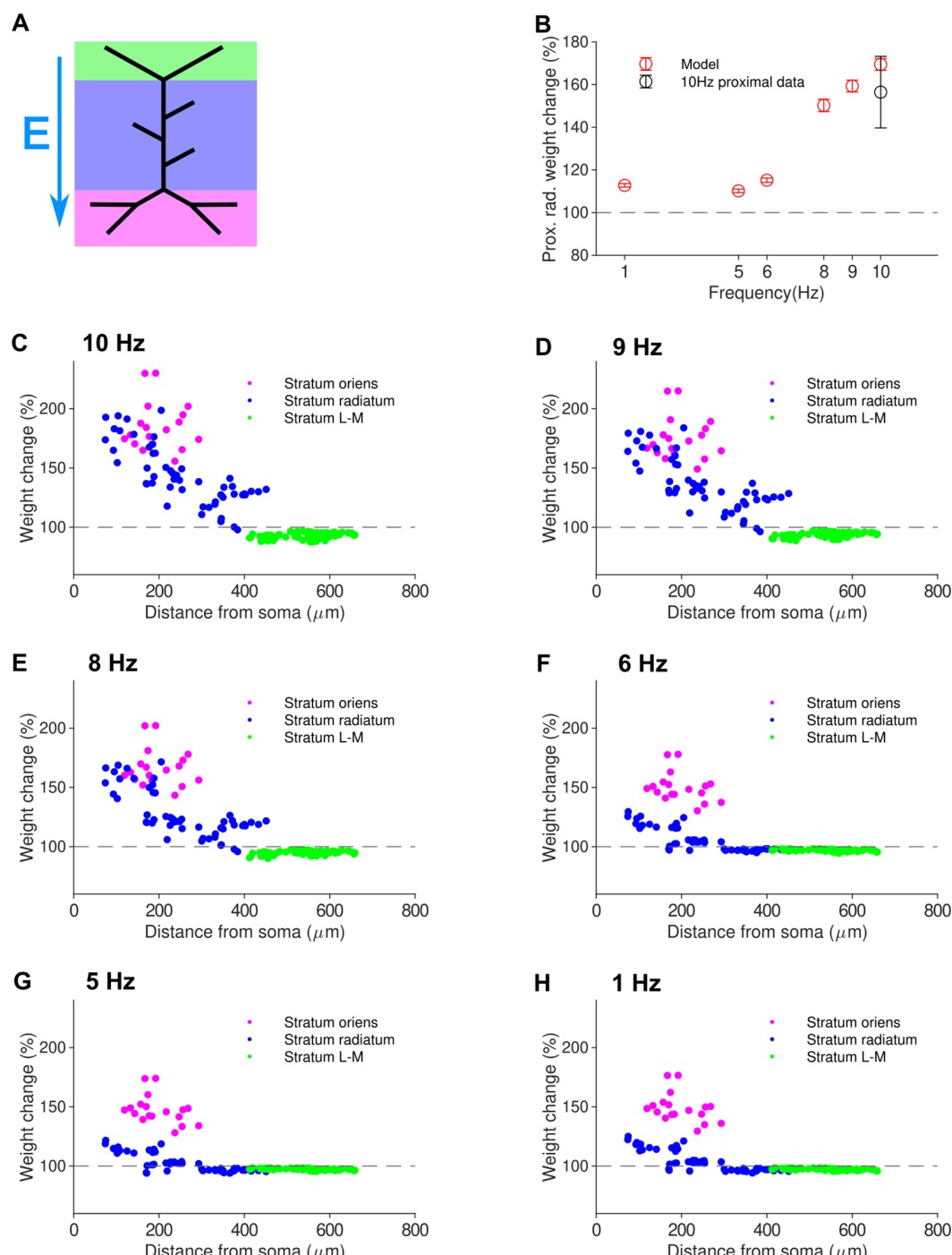

**Fig 3**. **The modelling framework for synaptic plasticity replicates proximal LTP induced by 10 Hz rMS protocol and predicts increasing LTP with higher stimulation frequency.** *(a)* Schematic of stimulation with electric field vector. *(b)* Mean LTP in the proximal stratum radiatum decreases with lower stimulation frequency; the induced LTP for 10 Hz rMS is in agreement with experimental data from Lenz (2015). *(c) - (h)* Induction of LTP from 900 pulse rMS protocol at various frequencies. Strong LTP is only seen in the proximal dendrites, with LTP in the stratum radiatum rapidly decreasing with lower frequency.

10 Hz local electrical stimulation generally does not induce LTP ([38]). Moreover, again in agreement with experiments [31], our simulations show that LTP induced by 10 Hz rMS is limited to proximal dendrites.

We also examined the dynamics of weight change over the time course of the simulation for a 900 pulse 10 Hz stimulus (Fig 4). For all dendritic locations, the weight change was close to monotonically linear, however different groupings of synapses experienced different rates of LTP/LTD.

To determine what the effect of higher frequencies on magnetic-stimulation induced LTP was, we implemented similar 900-pulse protocols at 15 and 20 Hz. We found that higher frequencies than 10 Hz continue to increase the amount of proximal LTP induced: 15 Hz caused mean proximal weight increase of 147%, and 20 Hz caused a 270% increase, which is a close to linear increase with increasing frequency (S3 Fig).

In additional simulations, we tested whether using biphasic instead of monophasic pulses caused different effects. We found that a biphasic protocol elicited substantially less LTP than monophasic protocols with identical other parameters (S1 Fig).

We also investigated the effect of magnetic stimulation intensity (i.e. the amplitude of the stimulation waveform). Increasing stimulation intensity increased the amount of LTP elicited, however these increases became marginal above 300 V/m, and no LTP was shown below the cell's somatic firing threshold of 210 V/m. (S2 Fig).

Tonic inhibition (mediated by extrasynaptic GABA-A receptors) is known to affect the excitability of neurons and synaptic plasticity. As our synaptic plasticity model was tuned using 19 - 25 days old rats [38], some of which would not have developed tonic inhibition, we decided to test whether our main result of proximal 10 Hz-rMS-induced synaptic plasticity is robust for different levels of tonic inhibition. To assess this, we implemented a tonic GABAergic current (as used before, [46]) in the CA1 pyramidal cell model at three different levels, corresponding to total tonic inhibitory currents between 20 and 40 pA [47] Importantly, the distance (as well as frequency) dependence of LTP were preserved, but higher tonic inhibition steadily reduced the absolute synaptic weight changes (S4 Fig). This effect was primarily due to tonic inhibition suppressing a subset of rMS-evoked spikes and increasing the attenuation of back-propagating action potentials. In conclusion, the simulated 10 Hz rMS protocol reliably induced proximal LTP even in the presence of tonic inhibition but its size depended on the strength of tonic inhibition.

## Dependence of proximal rMS-induced LTP on dendritic sodium and synaptic NMDA channels

In the Lenz et al. study [31], the effects of selectively blocking certain ion channels and receptors on rMS-induced LTP were assessed. Drugs known to reduce the magnitude of rMS-induced LTP [31] include the sodium channel blocker TTX,

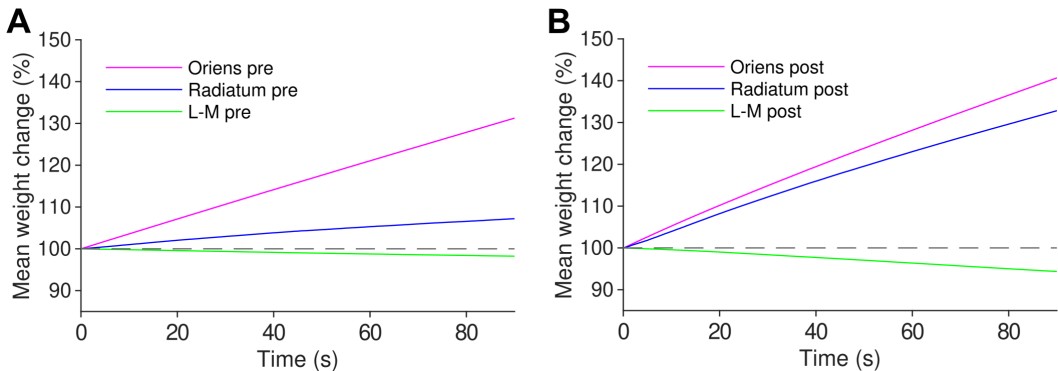

**Fig 4**. **Presynaptic and postsynaptic weights change monotonically linearly over stimulus duration.** *(a)* Time course of average presynaptic weights over stimulus duration for 900 pulse 10 Hz stimulus. *(b)* Time course of average postsynaptic weights over stimulus duration 900 pulse 10 Hz stimulus. Weight change in the stratum radiatum is far more postsynaptically dominated than in other strata.

L-type calcium channel antagonists and general calcium channel antagonists, as well as NMDA receptor antagonists, since both NMDA channels and voltage-gated calcium channels contribute to overall rMS-evoked LTP [31]. We thus tested the effects of a selection of *in silico* pharmacological perturbations on proximal LTP induced by 10 Hz of 900 rMS (Fig 5).

Importantly, in agreement with the experimental data [31], *in silico* suppression (see Methods) of dendritic sodium channels (simulating local TTX application) significantly reduced the amplitude of rMS-induced LTP at all frequencies as compared to control (Fig 5b). This indicates that our voltage-dependent plasticity rule correctly captures the dependence of proximal rMS-induced synaptic potentiation on sodium channel-mediated depolarisation. In fact, all sodium channels, including somatic ones, were blocked in the experiments of Lenz et al. Our simulations indicate that blocking dendritic sodium channels by local application of TTX would be sufficient to suppress the rMS-induced synaptic potentiation while keeping axosomatic generation of action potentials intact. Similarly, deactivation of NMDA receptors abolished all synaptic plasticity in the simulations (Fig 5c). Note that blocking NMDA receptors (and currents) does change baseline (i.e. pre-LTP) synaptic conductance but other manipulations (blocking inhibition, sodium and calcium channels or reducing extracellular calcium) do not.

We also tested *in silico* (see Methods) the suppression of L-type calcium channels and the use of a completely calcium-free solution, as these pharmacological perturbations have been reported to reduce rMS-evoked plasticity induction [31]. Suppression of L-type channels caused a large reduction in the amount of induced LTP, although not a complete abolition, as would be expected from Lenz et al. (Fig 5d). A calcium-free solution completely abolished any plasticity (Fig 5e).

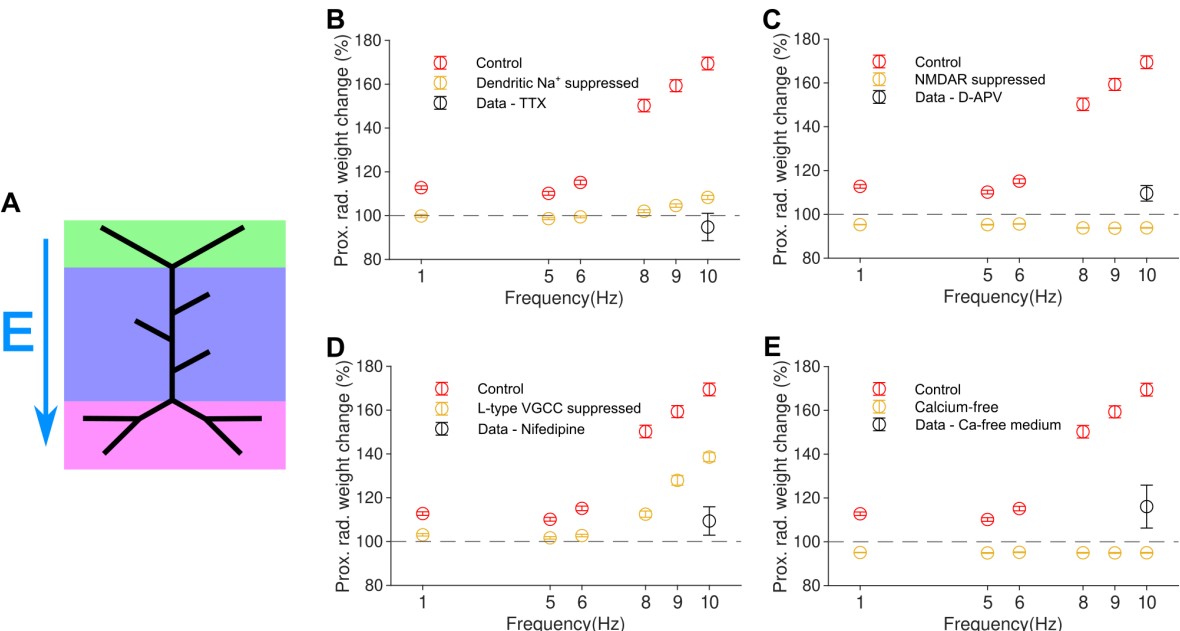

**Fig 5**. **Pharmacology *in silico*: rMS-induced LTP depends on dendritic sodium and calcium through voltage-gated calcium channels, as well as synaptic NMDA channels.** *(a)*: Schematic of electric field for all panels in this figure. Electric field vector direction shown by E. In addition, all regions of the CA1 pyramidal cell are synaptically stimulated because we assume that electric field not only depolarises the postsynaptic neuron but also elicits spikes in presynaptic neurons. *(b)*: Elimination of LTP when dendritic sodium channels are suppressed, simulating TTX application. *(c)*: Elimination of LTP when NMDAR (NMDA receptor)-like processes within the synaptic model are disabled *in silico*. *(d)*: Large reduction in LTP induction (as compared to control, i.e. no perturbation condition) when L-type voltage-gated calcium channels are suppressed *in silico*. *(e)*: Elimination of LTP observed when calcium-free solution is modelled.

Although we were unable to precisely match the final weight changes shown in experiment in [31], all of these interventions (which were shown in the paper of Lenz et al. to result in the stimulus causing no significant change to weights) did result in a substantial reduction of LTP induction or elimination of LTP entirely. Taken together, these data-driven pharmacological perturbations *in silico* demonstrate that our modelling framework allows for comprehensive mechanistic simulations of TMS-induced plasticity.

## Simulations predict distal LTP for theta burst stimulation protocols

Our model has so far shown that 10 Hz rMS elicits LTP localised in proximal dendritic locations. We wondered under what conditions LTP could be induced in the distal dendrites, particularly in the apical tuft of CA1 pyramidal cells. We investigated whether it was possible to induce LTP in the apical tuft, where synapses are typically targeted by axons originating in the perforant path from the entorhinal cortex.

The induction of LTP in these synapses has been shown to require the presence of dendritic spikes [48], mediated by active sodium channels in the dendrites. As the unifying plasticity model of Ebner et al. [30] is capable of generating LTP in response to local depolarizing events, we investigated whether similar local synaptic plasticity outcomes could be produced by rMS.

First, to replicate the experiments of Kim et al. [48], we applied their local electrical stimulation protocol of 15 bursts of 5 pulses, with each burst delivered at 100 Hz. In these simulations, no inhibition was present because, as in the experiments, to observe LTP in response to perforant path theta burst stimulation, GABA-A receptors had to be blocked pharmacologically [48]. We also neglected the inter-train interval present in the protocols of Kim et al. because our test simulations showed that inter-train intervals had a negligible effect on synaptic plasticity in this model (as we would expect in a model lacking metaplasticity, where inter-train intervals are longer than the time required for the synaptic model to reset), and would only increase the computational time required. In Kim et al.'s experiments, the soma was prevented from spiking. In our simulations, we did not implement any such clamp on the cell, since we found that during TBS, the somatic voltage depolarizations were so so small as to be insignificant, on the order of 2 mV, which is far below that which would generate a somatic spike.

As a comparison to these local, purely electrical stimuli, we investigated the effect of rMS delivered in a theta-burst fashion. In these cases, the rMS-like stimulus was a biphasic pulse delivered at 275 V/m. To account for the fact that rMS would stimulate many presynaptic neurons, synapses in all layers were activated (as in the 10 Hz rMS simulations above), not just those in the perforant path.

We compared the LTP/LTD distance profiles of all four cases - electrically induced TBS with and without inhibition, and rMS TBS with and without inhibition (Fig 6). With local purely electrically induced TBS and no inhibition, LTP was observed in the distal tuft, localised at the tips (Fig 6c). LTP was greatly reduced in the presence of inhibition (Fig 6b). In contrast to local electrical TBS, we observed LTP not only in proximal branches but also in the distal tuft under an rMS-TBS protocol, both in the presence and absence of inhibition (Fig 6e, 6f). Similar to the 10 Hz rMS protocol, LTP induction was also observed in more proximal synapses, to a similar extent as in the distal tuft. In summary, our simulations reproduce and predict the distal strengthening of synapses by electric and rMS TBS stimulation, respectively.

We observed the induction of dendritic spikes in the distal tuft when local electrical stimulation was delivered to the perforant path synapses in stratum lacunosum-moleculare. Similarly, we observed large-amplitude voltage depolarizations composed of EPSPs, bAPs and presumably also dendritic spikes, when rMS was delivered to the entire cell (Figs 7b, 7d). Inhibition suppressed these dendritic voltage events with local electrical stimulation of the perforant path, but not with rMS (Figs 7f, 7h). These somatic and dendritic voltage traces show that local depolarisations that exceed local plasticity thresholds in the dendrite can mechanistically explain why local electric TBS induces distal LTP, whereas rMS TBS induces proximal and distal LTP that is less sensitive to synaptic inhibition.

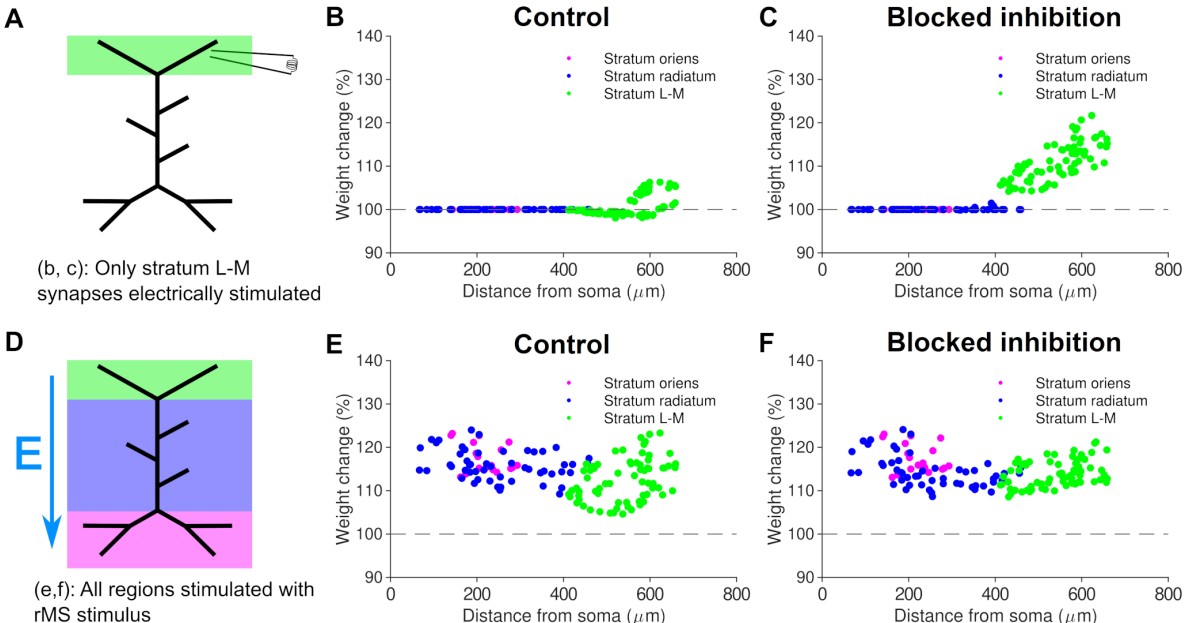

**Fig 6**. **The synaptic plasticity model reproduces distal LTP for local electric TBS and predicts proximal and distal LTP for rMS TBS.** Induction of LTP by local electrical TBS to the perforant path. *(a)*: Schematic of local electric stimulation for panels b and c. L-M: lacunosum-moleculare (perforant path target). *(b)*: Negligible plasticity observed when inhibition is present. *(c)*: Distal LTP observed when inhibition is blocked. *(d)*: Schematic of rMS simulation with electric field vector for panels e and f. *(e)*: LTP in both distal and proximal synapses when rMS is used. Unlike local electrical TBS, rMS TBS is able to induce large LTP when inhibition is present. *(f)*: LTP in both distal and proximal synapses when rMS is simulated and inhibition is blocked.

## Simulated distal LTP evoked by rMS TBS is dependent on sodium dendritic spikes and less sensitive to inhibition than local electrical TBS

Using local application of TTX, experiments by Kim et al. showed that distal LTP is dependent on sodium dendritic spikes [48]. Our model was able to reproduce this result (Fig 8) providing further validation of the plasticity model. We eliminated dendritic spikes by simulating an application of TTX to the apical dendrites, which deactivates the sodium channels in the apical dendrites but not in the soma.

In the absence of inhibition (bicuculline condition), local electrical TBS was able to restore LTP, whereas rMS TBS induced a slightly greater LTP amplitude as compared to the control condition. In contrast to blocking inhibition, the simulated application of TTX abolished LTP under local electrical TBS and also abolished LTP under rMS TBS.

These results show that our model is capable of producing consistent LTP in the distal tuft of pyramidal cells when local electrical or rMS theta-burst stimuli are applied. This represents a successful reproduction of experimental observations in the case of local electrical TBS. In addition, our simulations predict powerful LTP when using the rMS version of the TBS protocol instead. Furthermore, the model predicts that this distal LTP will be less diminished by the presence of inhibition in the case of rMS than in the case of local electrical stimulation. However, in both cases, the presence of dendritic sodium spikes is critical for the induction of LTP.

## Discussion

In this work, we have implemented a robust voltage-dependent modelling framework for long-term synaptic plasticity in neurons with extended dendrites. This framework goes beyond point neuron [49] and mean field modelling approaches [50–53] by allowing the simulation of layer and location dependent plasticity effects. Our plasticity simulations in the

**A** **Electrical TBS, inhibition blocked: Soma**

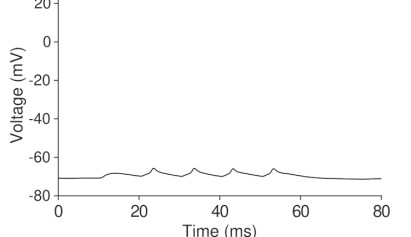

**B** **Electrical TBS, inhibition blocked: Tuft**

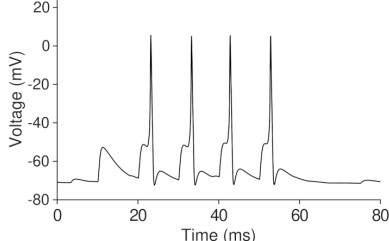

**C** **rMS-TBS, inhibition blocked: Soma**

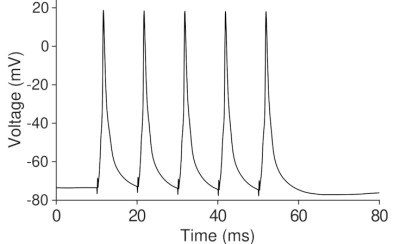

**D** **rMS-TBS, inhibition blocked: Tuft**

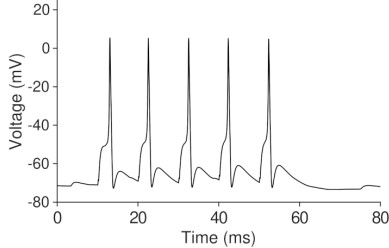

**E** **Electrical TBS, control: Soma**

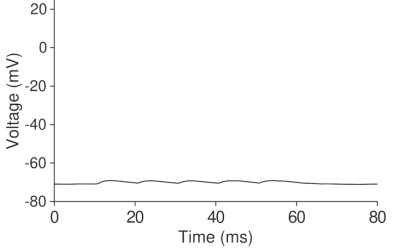

**F** **Electrical TBS, control: Tuft**

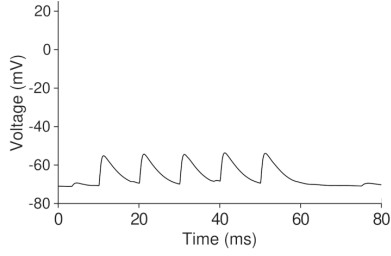

**G** **rMS-TBS, control: Soma**

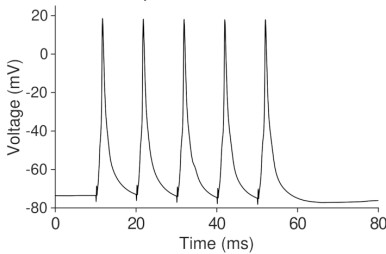

**H** **rMS-TBS, control: Tuft**

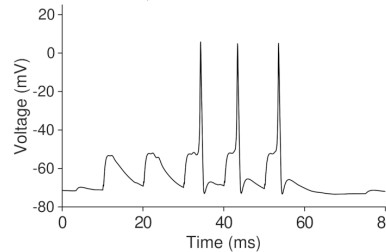

**Fig 7**. **Voltage traces from local electrical and rMS TBS, recorded at the soma and distal apical tuft, account for the outcomes of synaptic plasticity.** *(a)* 100 Hz local electrical TBS only causes small somatic voltage depolarisations when inhibition is blocked. *(b)* 100 Hz local electrical TBS induces dendritic spikes when inhibition is blocked. *(c)* 100 Hz rMS TBS causes multiple somatic spikes. *(d)* 100 Hz rMS TBS induces dendritic large-amplitude voltage depolarisations when inhibition is blocked. *(e)* 100 Hz local electrical TBS causes almost no somatic voltage depolarisation when inhibition is enabled. *(f)* 100 Hz local electrical TBS does not induce dendritic spikes when inhibition is enabled. *(g)* 100 Hz rMS TBS causes multiple somatic spikes with inhibition enabled. *(h)* 100 Hz rMS TBS induces dendritic large-amplitude depolarisations even when inhibition is enabled.

*NeMo-TMS* toolbox have provided a better mechanistic and quantitative understanding of experimental results on proximal LTP induced by 10 Hz rMS, including pharmacological perturbations using blockers of synaptic and intrinsic ion channels. We made specific predictions for long-term synaptic plasticity induced by lower frequencies of rMS. In addition, we reproduced experimental data on distal LTP induced by local electrical TBS of perforant path synapses and made

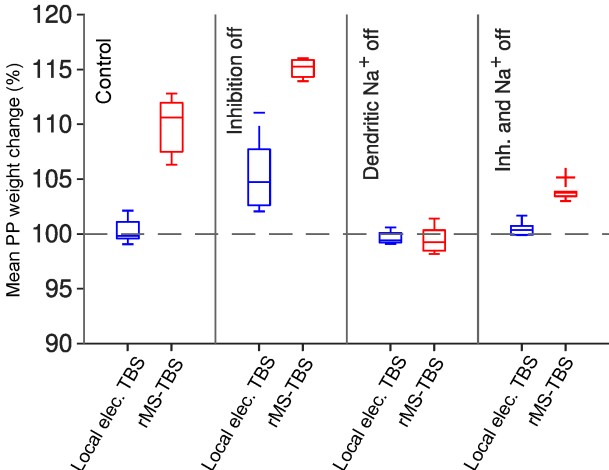

**Fig 8**. **Distal LTP induced by simulated rMS is sustained by sodium dendritic spikes but less affected by inhibition than local electrical stimulation.** rMS TBS (red) generates LTP even when local electrical TBS (blue) does not. In the presence of simulated GABAergic inhibition, rMS induces LTP comparable to local electrical stimulation without inhibition (i.e. bicuculline simulation). When dendritic sodium spikes are blocked (i.e. when local dendritic TTX application is simulated), neither of the two stimulation protocols we tested was able to induce LTP in the dendritic tuft. We also tested blocking somatic as well as dendritic sodium channels, and found effectively identical results (not pictured). Sample of 10 models with different randomised synapse locations.

predictions for proximal and distal LTP induced by rMS TBS, again including pharmacological perturbations. Moreover, our simulations indicate that 1 Hz rMS does not induce long-term depression.

## Modelling TMS-induced synaptic plasticity at subcellular resolution

rMS has previously been shown to induce LTP in a subset of proximal synapses [31]. However, there have been no computational tools to simulate and investigate layer- or compartment-specific rMS-induced synaptic plasticity at a spatio-temporal level with sufficiently high resolution. By incorporating our detailed synaptic plasticity model into the *NeMo-TMS*, it is now possible to simulate TMS-induced plasticity effects at selected spatial and temporal scales. Previously published tools used neural field theory and STDP modelling [54] to predict long-term synaptic changes in populations of neurons driven by repetitive and paired-pulse TMS protocols [50–53]. This type of modelling allowed estimation of the burst rates and number of pulses per burst that result in synaptic weight changes. However, because neurons were not represented as individual cells with dendrites and axons, but as averaged groups of neurons, layer-specific or compartment specific changes of synaptic strength could not be modelled (see also [49]). By combining multi-compartmental neuron modelling with a voltage-based plasticity rule, our work has filled this gap. In addition, while the basic version of *NeMo-TMS* [34] included a very simple model of synaptic activation (one strong synapse placed in the apical trunk), our current version of *NeMo-TMS* allows arbitrary placement of many synapses, with customisable parameters to determine their distribution within the dendrites of neuronal models. This allows for the investigation of location-dependent dynamics and plasticity of synaptic inputs in neuronal somata and dendrites.

## Modelling distance-dependent gradients of TMS-induced plasticity in neuronal dendrites

In agreement with experiments [31], our simulations showed strongly distance-dependent synaptic changes induced by rMS, with robust LTP in the proximal synapses. Our voltage-dependent plasticity model indicates that proximal LTP can

be explained by stronger proximal depolarisation evoked by rMS-evoked back-propagating axonal spikes and their coincidence with proximal postsynaptic responses as compared to distal postsynaptic responses.

Based on the observation of strongly distance-dependent LTP, we asked under what conditions TMS could induce potentiation in distal synapses. We made the prediction that an rMS TBS applied to all synapses and the postsynaptic neuron would generate LTP not only in proximal but also in distal synapses. These predictions can be tested in organotypic hippocampal slice cultures. In these TBS simulations, MS caused the cell to fire in response to almost every stimulus pulse, with little difference with or without inhibition, so the depolarisation at the distal tuft was very similar with or without inhibition. Therefore, because of somatic spike induction, magnetic TBS was able to induce LTP in these distal synapses, unlike local electrical TBS when LTP was blocked by inhibition. However, in the case of TTX simulation (local blockade of dendritic sodium channels), LTP was prevented even with rMS TBS, because inhibition was able to attenuate the back-propagating action potentials from somatic firing, with the backpropagation unable to amplify itself in the absence of sodium channels.

## Assumptions about synaptic activity, magnetic fields and neuronal morphology

In this modelling framework, we made a number of physiologically plausible assumptions in both the synaptic model and the presynaptic stimulation. It is known that cells in organotypic slice cultures are not silent [55], so we added 3 Hz Poisson background activity to all excitatory and inhibitory synapses. Multielectrode recordings or calcium imaging could be used to better characterise the ongoing neuronal activity in the entorhinal cortex and the CA3 of organotypic slice cultures. Such data could be used to improve the simulations of background synaptic activity.

Similarly, in our simulations, the presynaptic stimulation from both local electric and magnetic stimulation was delivered synchronously to all synapses with a constant delay of 1 ms. In the case of rMS, we believe this is a partially valid assumption because we used a uniform electric field. We assumed that this would have resulted in simultaneous activation of a large proportion of incoming axon terminals. We therefore modelled a simultaneous activation of all synapses, both excitatory and inhibitory. To make these simulations more realistic, random synaptic failures and stochasticity in synaptic delays could be included in the future, based on experimental data.

Furthermore, in our simulations, we imposed the stimulation as a uniform magnetic stimulus. This is a reasonable approximation for simulating neurons in an in vitro environment. However, replicating these results in a more realistic setting including an explicit dish model of the slice culture [56] or a head model and a non-uniform electric field would be an interesting further development due to the spatial heterogeneity and variability of magnetic and electric fields.

Since spatial effects are significant in the simulation of magnetic stimulation, simplified point neurons are not appropriate. On the other hand, detailed full-morphology models are computationally expensive for tuning and running long-term plasticity simulations [43]. Therefore, as a compromise we used a model of pyramidal CA1 neuron model with simplified dendrite branching and layers. While this morphologically reduced model of the CA1 pyramidal cell closely replicates key five dendritic and somatic electrophysiological features [37] of the actual cell (see Methods), including non-linear dendritic integration and backpropagation of action potentials [35], its dendrites have reduced branching and less realistic diameters as compared to full morphological models. Therefore, applying the plasticity modelling framework to morphologies with full dendrites of realistic diameter and branching may improve simulations of local dendritic voltage-based plasticity, which has been observed in experiments [57]. An additional caveat is that the CA1 pyramidal cell model coupled to the synaptic plasticity rule was fitted to hippocampal acute slice data [38] and later used to replicate experiments in hippocampal organotypic slice cultures [31]. Although the model has been used to reproduce several additional hippocampal datasets [48], future modelling should be improved by using a CA1 pyramidal cell model specifically developed and tuned to patch-clamp electrophysiology from organotypic slice cultures.

## Metaplasticity, inhibitory plasticity, and clinically relevant stimulation protocols

Metaplasticity is a phenomenon where the history of a neuron's activity causes persistent shifts in the threshold of LTP/LTD induction. Such shifts may occur due to activation of NMDARs or metabotropic glutamate receptors [58] or other biological mechanisms, which are currently unaccounted for in our model. As a consequence, weights change in a linear manner during stimulation (see Fig 4) and are only limited by hard bounds in our implementation. To increase biological plausibility, metaplasticity could be introduced by applying frameworks such as the BCM (Bienenstock-Cooper-Munro) [59] theory, in which the threshold for LTP/LTD induction is dynamically adjusted depending on neuronal activity. For example, a BCM-like model based on metaplastic STDP [60,61] could be integrated with the current four-pathway voltage-dependent plasticity model to provide a more realistic framework for the prediction of plasticity dynamics.

Including metaplasticity would also be useful for simulating the effect of inter-train intervals [50,53], which we neglect in our TBS simulations. Although originally included in rTMS protocols to avoid overheating the device, they have some effect on plasticity induction [62]. Metaplasticity would also be particularly relevant for investigating the "priming" of inhibitory theta burst stimulation by repetitive magnetic stimulation, where a low-frequency magnetic stimulus is used to prepare larger magnitudes of LTP induction with a subsequent potentiating stimulus.[63]

Furthermore, given that the presence or absence of inhibition has a modulatory effect on long-term synaptic plasticity [64–66], a more detailed modelling of diverse inhibitory mechanisms would be welcome. Here, we simulated inhibition as being activated simultaneously with excitatory inputs, but simulating feed-forward and feedback inhibition would be more realistic, as the E-I balance in CA1 pyramidal cells depends on complex interactions between different inhibitory cells [67–70].

As we used non-plastic inhibitory synapses, we also neglected possible contributions of inhibitory plasticity [71–74]. It has been observed in the hippocampus that rMS is able to reduce dendritic but not somatic synaptic inhibition [75]. *NeMo-TMS* may be particularly well suited to simulate such compartment-specific changes of GABAergic synapses. Long-term potentiation facilitated by disinhibition from local interneurons is a topic of active research [76], so a model of rMS that releases the inhibitory brake on excitatory plasticity [77] would be desirable. Importantly, inhibitory plasticity is capable of changing the sign of excitatory plasticity [71] and affecting location-specific E-I balance [78]).

With the addition of metaplasticity, inhibitory plasticity, or both, biologically realistic modelling of more complicated clinically relevant stimulation protocols such as intermittent theta burst stimulation (iTBS) [6,79,80] or continuous theta burst stimulation [81] could be implemented (cf. [50,53,82]).

## Conclusions

We have developed a new modelling framework for biophysically detailed simulations of TMS-induced synaptic plasticity and validated it in a well-established model of the CA1 pyramidal cell. We find that our framework is able to reproduce the rMS-induced LTP in the CA1 pyramidal cell model at frequencies well below those that would induce LTP in a local electrical stimulation regime. The rMS-induced synaptic plasticity is distance dependent, occurring very strongly in the proximal dendrites, and also frequency dependent, with stronger LTP observed at 10 Hz and 5 Hz compared to 1 Hz. We predict that magnetic TBS induces LTP in both proximal and distal dendrites of CA1 pyramidal cells.

Extending our modelling framework from the single neuron to the large circuit level and from animal to human synaptic data (e.g. [83]) will open new possibilities for predicting clinically relevant plasticity outcomes of rTMS.

## Supporting information

**S1 Fig. Biphasic stimulus shows different frequency dependence to monophasic stimulus**. Biphasic stimulus produces much less LTP than monophasic stimulus for 9 and 10 Hz cases, but there is no noticeable difference for lower frequencies.
(TIFF)

**S2 Fig. LTP induced by 10 Hz rMS rapidly increases after crossing firing threshold, but quickly saturates.** Below the firing threshold of 210 V/m, no LTP is observed, whereas once over the firing threshold, LTP amplitude increases, plateauing with intensities over 300 V/m.
(TIFF)

**S3 Fig. Plasticity model shows continued linear increases in LTP induction at higher frequencies.** At higher frequencies than 10 Hz, proximal str. radiatum weights increase by greater amounts, in a linear relationship with increasing frequency.
(TIFF)

**S4 Fig. Tonic inhibition reduces LTP amplitude but maintains proximal potentiation and frequency dependence.** Effect of tonic inhibition on distance and frequency dependence of long-term potentiation. (a–c): Induction of LTP with 10 Hz 900 pulse protocol with varying tonic inhibition conductances ($0.5-1 \times 10^{-4}$ $S/cm^2$, corresponding to 20–40 pA). (d) Frequency dependence of proximal LTP induction with different tonic inhibition conductances from variable frequency 900 pulse protocol.
(TIFF)

**S5 Fig. Plasticity model reproduces an additional synaptic plasticity paradigm using pairings of presynaptic spikes or bursts of spikes with postsnyaptic spikes or bursts of spikes.** LTP is not induced by single paired postsynaptic stimulus (cases 1/1 and 3/1 for single presynaptic or 3x presynaptic paired with 1x postsynaptic) but is produced by burst postsynaptic stimulus (cases 1/3 and 3/3 for single or 3x presynaptic paired with 3x postsynaptic).
(TIFF)

**S6 Fig. TMS pulse propagates beginning in the axon terminal, subsequently activating soma and dendrites.** Propagation of an action potential induced by a 250 V/m monophasic TMS pulse. (a): Voltage trace at various recording locations showing initiation of action potential at axon terminal (b): Schematic of cell showing the recording locations.
(TIFF)

**S7 Fig. Schematic of added myelinated axon.** Same color segments have identical biophysics. Blue: Axon initial segment and nodes; Green: Myelin; Red: Unmyelinated terminal.
(TIFF)

**S8 Fig. HippoUnit test error scores show the validity of the CA1 pyramidal cell model.** Error scores from HippoUnit test suite for 1. somatic spiking features, 2. depolarisation block, 3. back- propagating action potentials (bAPs), 4. attenuation of EPSPs, 5. nonlinear synaptic integration in oblique branches. Red line represents the acceptability threshold of two standard deviations. The model performed within two standard deviations of the experimental data and was considered acceptable.
(TIFF)

## Author contributions

**Conceptualization:** Nicholas Hananeia, Christian Ebner, Christos Galanis, Hermann Cuntz, Alexander Opitz, Andreas Vlachos, Peter Jedlicka.

**Data curation:** Nicholas Hananeia.

**Funding acquisition:** Alexander Opitz, Andreas Vlachos, Peter Jedlicka.

**Investigation:** Nicholas Hananeia, Christian Ebner, Christos Galanis.

**Methodology:** Nicholas Hananeia, Christian Ebner.

**Software:** Nicholas Hananeia.

**Supervision:** Peter Jedlicka.

**Visualization:** Nicholas Hananeia.

**Writing – original draft:** Nicholas Hananeia, Peter Jedlicka.

**Writing – review & editing:** Nicholas Hananeia, Hermann Cuntz, Alexander Opitz, Andreas Vlachos, Peter Jedlicka.

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
