## [Decision Letter · Decision Letter 0]

29 Oct 2024

PCOMPBIOL-D-24-01103Multi-scale modelling of location- and frequency-dependent synaptic plasticity induced by transcranial magnetic stimulation in the dendrites of pyramidal neuronsPLOS Computational Biology Dear Dr. Nicholas Hananeia, Thank you for submitting your manuscript to PLOS Computational Biology. After careful consideration, we feel that it has merit but does not meet PLOS Computational Biology's publication criteria as it currently stands. Therefore, we invite you to submit a revised version of the manuscript that addresses the points raised during the review process. Please submit your revised manuscript within 60 days. If you will need more time than this to complete your revisions, please reply to this message or contact the journal office at ploscompbiol@plos.org. Please include the following items when submitting your revised manuscript: * A rebuttal letter that responds to each point raised by the editor and reviewer(s). You should upload this letter as a separate file labeled 'Response to Reviewers'. This file does not need to include responses to formatting updates and technical items listed in the 'Journal Requirements' section below.* A marked-up copy of your manuscript that highlights changes made to the original version. You should upload this as a separate file labeled 'Revised Manuscript with Track Changes'.* An unmarked version of your revised paper without tracked changes. You should upload this as a separate file labeled 'Manuscript'. If you would like to make changes to your financial disclosure, competing interests statement, or data availability statement, please make these updates within the submission form at the time of resubmission. Guidelines for resubmitting your figure files are available below the reviewer comments at the end of this letter. We look forward to receiving your revised manuscript. Kind regards, Ausra Saudargiene, Ph.D.Academic EditorPLOS Computational Biology Andrea E. MartinSection EditorPLOS Computational Biology Feilim Mac GabhannEditor-in-ChiefPLOS Computational Biology Jason PapinEditor-in-ChiefPLOS Computational Biology  **Journal Requirements:** **Additional Editor Comments (if provided):****Reviewers' comments:** Reviewer's Responses to Questions

**Comments to the Authors:**

Reviewer #1: The idea behind the manuscript is very interesting: an in-silico framework for reproducing plasticity inducing paradigms utilizing magnetic stimulation. However, I do not think that this manuscript is suitable for publication in this form.

Major issues:

1. The manuscript reads like a draft that has not been edited or even ready by any of the senior authors. One can find many weird expressions (such as NMDAR supression: one inhibits a receptor/channel or supresses its activity) and jargon in the manuscript. The writing feels sloppy --- there are entire paragraphs (e.g. lines 49-57, 111-122) that are completely incomprehensible.

2. The link to the model allows only to download a part of it. The reviewer is unable to test if the code runs on their machine.

3. The methods section is insufficient. The model neuron is not sufficiently described. The dataset it has been fitted to is not described. The synaptic plasticity model is not described at all. There are many instances of including technical details that were not described in the methods section e.g. line 134 a model variable N_alpha that the reader knows is set to 0. It is also impossible to look at the model code to check oneself what are introduced changes.

3a. There are no figures showing validation of the reduced model of the CA1 cell. Or even a value number describing validity of the fit. The dataset the cell was fitted to is not correctly described.

4. There is no justification provided for using a model of pyramidal CA1 neuron fitted to slice data to replicate experiments with organotypic hippocampal cultures. There is no discussion on the validity and limitations of this approach.

5. Tonic inhibition. In addition to inhibitory synapses (phasic inhibition) there are also extrasynaptic GABAA receptors that excert a strong inhibitory influence, usually reducing cell's excitability. The dataset used to fitting utilized mice older than 3 weeks with ACSF not including GABAAR inhibitors (not blocking tonic GABA inhibition, which is developing/present), so the model neuron implicitly includes this tonic inhibition. In validation of the synaptic plasticity model using Ikegaya et al data, which was collected in 19–25 days old rats. Some of these rats would not have developed tonic inhibition. The in-silico experiments should address the question of higher excitability of neurons in Ikegaya et al. data. Blocking inhibition in TBS simulations should address the issue of tonic inhibition.

6. Please use more than one synaptic plasticity paradigm in addition to data from synaptic plasticity experiments utilizing organotypic slice cultures to fit and validate the synaptic plasticity model.

7. Fig. 3 to : Please include plots with measurable variables, so it is easier to compare model predictions with experimental data.

8. Discussion is lacking.

9.

Minor issues:

1. The title refers to modeling transcranial magnetic stimulation (TMS), which is actually not modeled in the manuscript. The authors reproduce in silico experiments with organotypic hippocampal cultures using magnetic stimulation. It is not a manuscript on modeling TMS.

2. Matlab is propriety, hence the in-silico experiment is not reproducible.

3. I would personally move all the tables from the supplement to the methods section to make it easier to read.

4. Fig2: Ikegaya et al. data were collected in slice. There might be a good reason for adding noise, but it is not reproducing the non silent nature of slice cultures.

5. There definitely is a better way to evaluate the fit than the statement in line 216. Why are there no errorbars for simulated data (line 76 states that there were 10 test models).

Minor minor issues:

line 74, 92 test battery -> test suite

line 102 Exp2Syn is probably a double exponential. Provide mathematical description.

line 168 I do not know what "behaved within acceptable electrophysiological limits for a CA1 pyramidal cell" means. Please provide a figure illustrating that.

line 223 simulates -> reproduces

lines 243 -- 247 I don't understand that paragraph

line 255 mechanistic basis?

line 279 Our model has so far shown that 10 Hz rMS produces LTP concentrated in proximal dendritic locations. -> elicits (...) localized in proximal dendrites

line 299 probably an effect of tonic inhibition.

Recommended literature:

[1] Groen MR, Paulsen O, Pérez-Garci E, Nevian T, Wortel J, Dekker MP, Mansvelder HD, van Ooyen A, Meredith RM (2014) Development of dendritic tonic GABAergic inhibition regulates excitability and plasticity in CA1 pyramidal neurons. J Neurophysiol 112:287–299. 10.1152/jn.00066.201

Reviewer #2: Hananeia et al. developed a model of CA1 pyramidal cell that integrates a plasticity model from their earlier work and describes the synaptic plasticity-inducing effects of transcranial magnetic stimulation. The authors show that their model reproduces observations on proximal synaptic plasticity and explore the possibility of using magnetic stimulation to induce plasticity in distal synapses. The work seems to be well conducted, but the results and methods should be described more clearly or in more detail.

Major concerns:

-The model is not properly described. I would suggest including at least a minimal description of all model components (especially the ones underlying the parameters listed in the tables) either in Methods or Supplementary material. The online model entry has issues too. I tried to use the model they uploaded in dropbox but faced issues. The function "t2n_initModelfolders" that is used by the simulation scripts is not defined anywhere. Also the scripts point to a folder ../Models which doesn't exist. The README file could be more informative - it tells me about the I/O needs (which is great) but not about the runtime of the simulations.

-The terminology related to the stimulus is somewhat confusing - at least if you're not completely familiar with the protocols used in this branch of neuroscience. From the Methods I understand that there are two stimuli: 1) The synaptic activation (Schaffer collateral synapses) and 2) a somatic current injection. In the experiments of Fig. 2, only the synaptic stimulation is mentioned in the main text, but the caption also mentions local electrical stimulation. What is this local electrical stimulation? Later it is termed "local electrical stimulation of hippocampal CA1 pyramidal cell synapses". If this is only synaptic stimulation, I would call it plainly "synaptic stimulation" or "Schaffer collateral stimulation" to distinguish from the somatic stimulation. However, in Fig. 2 I'm left wondering whether the somatic stimulation is in place or not. It is also unclear what is the main difference in the protocols between Fig. 3-4 and Fig.5: why is there no weight change for proximal synapses in Fig.5 while there was a strong potentiation in Fig.3-4?

-One problem with the terminology is that "rMS" ("repetitive magnetic stimulation") seems to occasionally used as if it was a term for a specific stimulation protocol. Please use another term if you refer to a specific protocol. Magnetic TBS is also repetitive magnetic stimulation. In particular, the manuscript might benefit from a section or table in the Methods where each protocol is explained.

-Please include one panel where the time course of the synaptic weight change is plotted for one of each synapse class (green, purple magenta) in Fig 3. Would be also useful in Fig. 2. It would also be beneficial for the reader to see panels that show what is the contribution of pre- vs. postsynaptic LTP/LTD to the reported LTP/LTD outcome, similar to Figure 1E in Ebner et al. 2019.

Minor comments:

-Methods: "To simulate dendritic TTX application, the conductances of all sodium channels in the dendrites of the cell were set to zero": Only dendritic Na+ channels or also somatic/axonal ones? What happens when all Na+ channels are blocked, are the results the same?

-Methods: "In addition to the rMS pulses, we applied a random asynchronous input at 3 Hz to mimic spontaneous neuronal activity in organotypic slice cultures.": You should say this is Poisson-type activity. On line 397 you mention it is "3 Hz background activity with random jitter" - this is also less informative than just saying that it's a 3-Hz Poisson process.

-Methods: The TMS waveform generator is not a standard method, at least to me. Please explain in the methods what the 275 V/m means in practice and how it is implemented in the neuron model. A brief pulse of electrical current applied in a distance-dependent manner to each compartment? I do not find anywhere information on how long these pulses are.

-Fig. 2: Green colour missing from (A). However, the color coding doesn't seem to be used at all in Fig. 2?

-Fig. 2: Are the simulation results (red circles) averages over 128 synapes? A minor point, but I would put the simulations with solid line and experimental data with separate points because you can easily make more simulated results and (supposedly) get a smooth curve while you are using experimental data as given in Ikegaya et al.

-Fig. 3: What drives the LTD in distal synapses in 10 Hz stimulation? Do the proximal LTP and distal LTD become stronger for increased frequencies?

-Fig. 4: I would suggest not to call the control experiments "baseline", since usually "baseline" in LTP/LTD terminology means the status of the system before the plasticity-inducing protocols.

-Fig. 2 and 4: Do the manipulations (blocking of inhibition, Na+ channels, NMDA currents, L-type Ca2+, or extracellular Ca2+) affect the baseline synaptic conductance?

-Discussion: There are two paragraphs about metaplasticity (which was not included in the model), which is a bit difficult to follow. The authors seem to equate metaplasticity modelling with modelling using plasticity rules that are able to take into account events further back in time. While this can be considered an important aspect of metaplasticity, it is not the only one. Whether plasticity occurs depends also on the state of neuromodulation and other internal states that may or may not be affected by the history of synaptic inputs and spiking responses. I would therefore suggest a change in discussion the limitations of the model that you start by discussing the biological phenomena that you cannot yet describe with the model and then suggest the additional components needed to capture them rather than the other way around.

Reviewer #3: This paper proposes a modeling framework targeting the effects of magnetic and electric stimulation on the behavior of neurons with specific focus on long-term synaptic plasticity. In its presented form, the model targets CA1 hippocampal pyramidal cells. Its main hallmarks are the representation of the cells by biologically plausible, but simplified, morphologies (somewhat halfway between point neurons and fully realistic compartment models) and the inclusion of plasticity for excitatory synapses (but not meta-plasticity, nor plasticity of inhibitory synapses).

The authors demonstrate the potential of their framework for the explanation of existing experimental results (e.g., LTP-like plasticity for 10 Hz rMS in at proximal locations), for the generation of predictions that could be tested in future experiments (e.g., on frequency dependency of rMS induced plasticity and on proximal and distal LTP induced by tMS TBS), and for the generation of insight into the mechanisms underlying certain experimental observations (e.g. for the stronger proximal LTP).

The framework appears to be a useful tool for studying the cellular underpinnings of brain stimulation induced effects as well as for the development of new protocols for, e.g., clinical purposes. It bears substantial potential for future extensions – many of these are discussed in detail by the authors themselves. The paper is well-written and reasonably accessible also to community of users outside the modeling community.

In consequence, I recommend the manuscript for publication. Some minor corrections might be applied:

1) Please make sure that acronyms are explained before first use – e.g. rMS is line 44.

2) Line 54 “better explain” – better than what?

3) Line 115: “… model IS shown …”

4) Line 174. The reference should be S5

5) Line 176 onwards: this first section of Results seems to belong to the Method section.

Moreover, I think it would further increase the accessibility and completeness of the paper, if the authors would, in a bit more detail, describe the mathematical and biophysical foundations of the used neuron model, including the way the induced electric field is coupled to the neuronal states. I understand that these details are given elsewhere, but at least a brief summary with the most important equations would be helpful.

**Have the authors made all data and (if applicable) computational code underlying the findings in their manuscript fully available?**

Reviewer #1: **No: **The dropbox link leads to one directory of the repository, which is the Generator. The directory with the model files (Model) is missing. Scripts for reproducing figures are written in Matlab, which requires the reviewer to use Matlab, which is not a freely available software.

Reviewer #2: **No: **The code is available through dropbox but it was not runnable

Reviewer #3: Yes

PLOS authors have the option to publish the peer review history of their article (what does this mean?). If published, this will include your full peer review and any attached files.

Reviewer #1: No

Reviewer #2: No

Reviewer #3: **Yes: **Thomas R. Knösche

 **Figure resubmission:**While revising your submission, please upload your figure files to the Preflight Analysis and Conversion Engine (PACE) digital diagnostic tool, https://pacev2.apexcovantage.com/. PACE helps ensure that figures meet PLOS requirements. To use PACE, you must first register as a user. Registration is free. Then, login and navigate to the UPLOAD tab, where you will find detailed instructions on how to use the tool. If you encounter any issues or have any questions when using PACE, please email PLOS at figures@plos.org. Please note that Supporting Information files do not need this step. If there are other versions of figure files still present in your submission file inventory at resubmission, please replace them with the PACE-processed versions. 
---

## [Decision Letter · Decision Letter 1]

7 Sep 2025

PCOMPBIOL-D-24-01103R1

Multi-scale modelling of location- and frequency-dependent synaptic plasticity induced by repetitive magnetic stimulation in the dendrites of pyramidal neurons

PLOS Computational Biology

Dear Dr. Hananeia,

Thank you for submitting your manuscript to PLOS Computational Biology. After careful consideration, we feel that it has merit but does not fully meet PLOS Computational Biology's publication criteria as it currently stands. Therefore, we invite you to submit a revised version of the manuscript that addresses the points raised during the review process.

Please submit your revised manuscript within 30 days Nov 07 2025 11:59PM. If you will need more time than this to complete your revisions, please reply to this message or contact the journal office at ploscompbiol@plos.org. Please include the following items when submitting your revised manuscript:

We look forward to receiving your revised manuscript.

Kind regards,

Suhita Nadkarni, Ph.D

Academic Editor

PLOS Computational Biology

Hugues Berry

Section Editor

PLOS Computational Biology

**Additional Editor Comments:**

All reviewers are positive for this version of the manuscript and find the revised manuscript much improved and nearly ready for publication. The remaining comments are minor and focus on clarifying language, improving figure labels, and ensuring code reproducibility.

Reviewer 1 primarily emphasizes the importance of code accessibility and reproducibility. They request steps to ensure the model can be run by users without MATLAB or on non-Windows systems, such as providing a key standalone .hoc file.

Reviewer 2 provides extensive minor textual, grammatical, and labeling suggestions to enhance clarity, precision, and consistency throughout the manuscript and figures. Key points include:

Correcting minor typos and grammatical errors and the using more precise scientific language, defining terms like "LTP measure" and "amplitudes".

We ask that you please carry out these final minor revisions as advised by the reviewers to ensure the highest clarity and reproducibility before the manuscript is accepted for publication.

**Journal Requirements:**

1) Some material included in your submission may be copyrighted. According to PLOSu2019s copyright policy, authors who use figures or other material (e.g., graphics, clipart, maps) from another author or copyright holder must demonstrate or obtain permission to publish this material under the Creative Commons Attribution 4.0 International (CC BY 4.0) License used by PLOS journals. Please closely review the details of PLOSu2019s copyright requirements here: PLOS Licenses and Copyright. If you need to request permissions from a copyright holder, you may use PLOS's Copyright Content Permission form.

Potential Copyright Issues:

i) Figures 1, 2A, and 6A. Please confirm whether you drew the images / clip-art within the figure panels by hand. If you did not draw the images, please provide (a) a link to the source of the images or icons and their license / terms of use; or (b) written permission from the copyright holder to publish the images or icons under our CC BY 4.0 license. Alternatively, you may replace the images with open source alternatives. See these open source resources you may use to replace images / clip-art:

2) Please note that your Data Availability Statement is currently missing the repository name, and the DOI/accession number of each dataset OR a direct link to access each dataset. If your manuscript is accepted for publication, you will be asked to provide these details on a very short timeline. We therefore suggest that you provide this information now, though we will not hold up the peer review process if you are unable.

3) Kindly revise your competing statement to align with the journal's style guidelines: 'The authors declare that there are no competing interests.'

**Reviewers' comments:**

Reviewer's Responses to Questions

**Comments to the Authors:**

Reviewer #1: The manuscript was thoroughly rewritten and is a pleasure to read (and

look at!).

Minor comments:

97 is -> are

probably all ion names should be in \mathrm mode

104 - 106 was modified -> consisted of

107 - what is MS?

190-191 - what about opening of L-type Ca channels in the spines due to the depolarization accompanying plasticity induction?

Fig 2 panel B "Inhibition enabled" -> maybe "Control"

Fig 2 panel C "Inhibition disabled" -> maybe "Blocked inhibition"

I would probably change Simulation to Model

286 strongest fit -> best fit

287 weakest fit -> worst fit

292 this -> a

314 What is the LTP measure? It needs to be defined.

Fig 3 panel B: Lenz et al (2015) is looking at aEPSC amplitude, it might be a good idea to compare EPSC amplitude instead of approx. weight.

Fig 3 caption decreasing with lower frequency -> decreasing with decreasing stimulation frequency

319 remove word "here"

320 classical electrical presynaptic stimulation -> commonly used presynaptic plasticity-inducing stimulation protocols

321 localized to -> limited to

321 the proximal dendrites -> proximal dendrites

325 In all cases -> For all dendritic locations

327 we experimented with - please rewrite

333-336 I do not understand this sentence, please rewrite

338 depolarisation of neurons -> maybe excitability?

373 something is missing in this sentence, plasticity induction maybe?

377 I think simulated is unnecessary

379 amplitudes of what?

Fig 5 panels (C-D) no perturbation -> control

panel C NMDA -> NMDAR

panel D L-Type Ca2+ suppressed -> L-type VGCC, add VGCC definition in the caption

401 -> Nice!

404 -> rewrite sentence

411 electrical TBS -> maybe electrically induced TBS

419 stimulation is missing from the sentence

Fig 6 caption inhibition is disabled -> inhibition is blocked

Fig 7 deflections -- depolarizations

472 specific changes of synapse -> specific changes of synaptic strength

475 very simple synaptic modelling -> included a very simplistic model of synaptic activation (?)

502 stimulus -> stimulation

Reviewer #2: The authors have revised the manuscript and it's almost ready for publishing in my opinion. The only issues I have left are related to the publication of the code and reproducibility.

I agree with Reviewer 1 that the work suffers from being coded in MATLAB. Just to make this revision, I had to re-initiate my MATLAB license, adding a cost to my university's software budget. The code is also using toolboxes (randsample.m), which adds additional costs. I know it's comfortable to use the software one has become acquainted with, but a longer term solution would be for the authors to start using Python in their future publications.

And please don't use white spaces in folder names.

The code is now made more readable with new README files compared to the first version. Sadly, I have to conclude that even after many approaches I cannot run the simulations on my Mac. It seems that the system is tuned for Windows, and it would require large modifications to make it runnable in Mac, and perhaps the same for Linux/Unix systems. When preparing the final code publication, I would suggest that the authors 1) try it on Linux/Unix/Mac machines, and provide information how to make it runnable there, or 2) provide a "default" hoc file that contains one typical simulation and is runnable without all the MATLAB code generation. With the hoc and mod (and morphology files) in place, anyone can run the model regardless of their operating system and regardless whether they have MATLAB installed or not. Then the work would be at least minimally reproducible for all. There are also options to save large simulation scripts into online databases other than Github and ModelDB. Currently, it seems to me that the crucial hoc files are being made by the MATLAB script, and thus if that script does not work I cannot run any simulations.

Other minor comments:

Mostly the synaptic changes in stratum radiatum are discussed, while the changes in stratum oriens are predicted to be larger (1.5x to 2.0x in stratum oriens compared to 1.1x to 1.5x in stratum radiatum in Fig. 3). Could you relate these predictions to some experimental data?

Fig. 4: Use seconds or minutes on x axis instead of ms*1e4

L.344 "Importantnly"

L.431: "experiments by Kim et al. showed that distal LTP is dependent on sodium dendritic spikes [48]". [48] seems to have two different protocols: strong, local TTX close to soma/proximal dendrites, and bath application of low concentration of TTX that effectively blocks the dendritic Nav channels. I suppose the authors model the latter. However, I would assume that also the somatic Nav channels are somewhat affected by this procedure - there is no mentioning of specifically targeting the dendrite in the bath application. In [48] they suggest that bath application of low-concentration of TTX effectively blocks the dendritic Nav channels because they are low in number, while the Nav channels in the soma are less affected due to their density. This discrepancy should be discussed in the Discussion - alternatively, you could show that a simultaneous partial blockade of somatic Nav channels gives reasonable output.

Reviewer #3: My (relatively minor) concerns from the previous review have been adressed satisfactorily. I endorse publication.

**Have the authors made all data and (if applicable) computational code underlying the findings in their manuscript fully available?**

Reviewer #1: Yes

Reviewer #2: None

Reviewer #3: Yes

PLOS authors have the option to publish the peer review history of their article (what does this mean?). If published, this will include your full peer review and any attached files.

Reviewer #1: No

Reviewer #2: No

Reviewer #3: **Yes: **Thomas R. Knösche

**Figure resubmission:**
---

## [Editor Report · Decision Letter 2]

11 Nov 2025

Dear Mr. Hananeia,

We are pleased to inform you that your manuscript 'Multi-scale modelling of location- and frequency-dependent synaptic plasticity induced by repetitive magnetic stimulation in the dendrites of pyramidal neurons' has been provisionally accepted for publication in PLOS Computational Biology.

Best regards,

Suhita Nadkarni, Ph.D

Academic Editor

PLOS Computational Biology

Hugues Berry

Section Editor

PLOS Computational Biology

---

## [Editor Report · Acceptance letter]

PCOMPBIOL-D-24-01103R2

Multi-scale modelling of location- and frequency-dependent synaptic plasticity induced by repetitive magnetic stimulation in the dendrites of pyramidal neurons

Dear Dr Hananeia,

I am pleased to inform you that your manuscript has been formally accepted for publication in PLOS Computational Biology. Your manuscript is now with our production department and you will be notified of the publication date in due course.

With kind regards,

Anita Estes
